# A Feature-Enhanced Small Object Detection Algorithm Based on Attention Mechanism

**DOI:** 10.3390/s25020589

**Published:** 2025-01-20

**Authors:** Zhe Quan, Jun Sun

**Affiliations:** School of Artificial Intelligence and Computer Science, Jiangnan University, Wuxi 214122, China; 6223111057@stu.jiangnan.edu.cn

**Keywords:** small object detection, attention mechanism, feature pyramid network, detection head, loss function

## Abstract

With the rapid development of AI algorithms and computational power, object recognition based on deep learning frameworks has become a major research direction in computer vision. UAVs equipped with object detection systems are increasingly used in fields like smart transportation, disaster warning, and emergency rescue. However, due to factors such as the environment, lighting, altitude, and angle, UAV images face challenges like small object sizes, high object density, and significant background interference, making object detection tasks difficult. To address these issues, we use YOLOv8s as the basic framework and introduce a multi-level feature fusion algorithm. Additionally, we design an attention mechanism that links distant pixels to improve small object feature extraction. To address missed detections and inaccurate localization, we replace the detection head with a dynamic head, allowing the model to route objects to the appropriate head for final output. We also introduce Slideloss to improve the model’s learning of difficult samples and ShapeIoU to better account for the shape and scale of bounding boxes. Experiments on datasets like VisDrone2019 show that our method improves accuracy by nearly 10% and recall by about 11% compared to the baseline. Additionally, on the AI-TODv1.5 dataset, our method improves the mAP50 from 38.8 to 45.2.

## 1. Introduction

As object detection technology advances, unmanned aerial vehicles (UAVs) are increasingly being applied in diverse areas including traffic monitoring [1], power inspections [2], crop analysis [3], and disaster relief. For instance, in the field of traffic monitoring, UAVs can perform aerial surveillance, unconstrained by road limitations, with advantages such as their high speed, high degree of freedom of movement, and wide field of view. However, most studies have focused on the analysis of ground-based monitoring images, and object detection algorithms for UAV-perspective images have not yet been sufficiently explored. The main reasons are the challenges presented by UAV-perspective images, including scale variation, class imbalance, uneven sample distribution, a high number of objects predominantly small in size, and the difficulty of balancing the high computational demands of high-resolution UAV images with the limited computing power of current low-power chips. Compared to ground-level natural images, UAV-perspective images often require high-resolution imagery to extract features effectively due to the large field of view, uneven lighting, and dense object distribution. This significantly increases the computational and memory requirements of object detection algorithms. Employing unoptimized general object detection algorithms requires models with extensive parameters and significant computational power to maintain detection accuracy, leading to prohibitive computational and memory expenses. Consequently, enhancing the performance of object detectors while operating under resource limitations is a key challenge in UAV image object detection.

Object detection is a fundamental technology in computer vision, designed to allow computers to identify and pinpoint objects in images. This process involves not only recognizing the objects but also determining their locations within the image and classifying them into specific categories. In the early 21st century, most object detection used traditional algorithms, which generally involved extracting candidate boxes, extracting features from these candidate boxes, and then classifying them using a classifier. However, the performance and stability of these methods were generally poor. As a result, deep learning methods are now widely employed for object detection. Since the introduction of AlexNet [4] in 2012, the success of deep neural networks has been proven, marking the beginning of an era dominated by deep learning-based detection techniques. In today’s computer vision domain, deep learning-based object detection methods are primarily categorized into two types: single-stage and two-stage object detection methods. The RCNN [5,6,7] series, which first combined deep learning with object detection, is a typical two-stage detector. It generates numerous region proposals in the image after feature extraction, performs a simple foreground–background judgment, and then feeds the proposed regions into the network for further classification and the refinement of the detection boxes. Although two-stage object detection has certain advantages in terms of accuracy, it has significant issues in model complexity and real-time performance. The single-stage object detection method, on the other hand, directly classifies and locates features extracted from the input image via neural networks, bypassing the candidate box generation stage. Despite the resulting problem of imbalanced positive and negative samples, the good detection speed and real-time performance of the single-stage object detection method have led to its widespread use in daily production and life. Representative single-stage object detection algorithms include SSD [8], YOLO [9], and RetinaNet [10]. Additionally, since ViT [11] directly applied the Transformer encoder to image feature extraction, achieving cross-domain migration from natural language processing to the vision field, it has provided a new approach to handling visual tasks. DETR [12] represents a bold attempt to combine Transformers with object detection, eliminating the need for predefined anchors in the image and the use of NMS for post-processing. As an end-to-end object detector, DETR has achieved impressive results in object detection tasks. However, DETR also has significant drawbacks, including high computational complexity and the need for large amounts of data and computational resources for training. Given the same computational costs, it does not have a significant advantage over some CNN networks. However, in some natural scenes, object scales vary, shapes are irregular, and UAV-based object detection frequently involves focusing on small objects. Consequently, the mainstream object detection algorithms discussed earlier are not directly suitable for UAV aerial detection tasks.

Recently, researchers have dedicated substantial efforts to enhancing the accuracy of detecting small objects [13], addressing this challenge through dataset augmentation methods, including segmenting images to enhance the representation of small objects in each segment and inserting small objects into different backgrounds. This approach increases the number of small object samples and improves detector performance. Deng et al. [14] introduced an ultra-high-resolution, extended feature pyramid network (FPN) tailored to small object detection, along with the FTT module, which enhances feature extraction while minimizing detail loss in the feature map. Gong et al. [15] suggested that while the top-down connections between adjacent layers in FPN benefit small object detection, they also present some limitations. They introduced a novel concept—fusion factor—to manage the information flow from deeper to shallower layers, thereby optimizing FPN for detecting small objects. Noh et al. [16] enhanced detection performance by utilizing high-resolution target features as supervisory signals and aligning the relative receptive fields of input and target features. Additionally, QueryDet [17] initially estimates the approximate positions of small objects on a low-resolution feature map and subsequently creates a sparse feature map from the high-resolution features of these estimated positions to achieve precise detection results. This approach maximizes the use of high-resolution feature maps while avoiding unnecessary computations in background regions. Regarding loss functions, since IoU [18] (Intersection over Union) is highly sensitive to bounding box deviation for small objects, Wang et al. [19] developed a new metric tailored to the characteristics of small targets. They reformulated the bounding box as a two-dimensional Gaussian distribution, converting the Intersection over Union (IoU) between predicted and ground truth boxes into a measure of similarity between the two distributions. This similarity was then normalized to create a new evaluation metric, NWD (Normalized Wasserstein Distance), which quantifies the similarity between the distributions.

Inspired by the aforementioned research, we propose a UAV-perspective-based object detection model. This model uses YOLOv8 as the backbone network and introduces modifications to its structure and modules to enhance detection performance without consuming too many resources, ensuring a real-time capability. This paper’s primary contributions are as follows:We replaced the PAN (Path Aggregation Network) in YOLOv8 with HSPAN to better fuse features at different scales. And we incorporated a feature-enhanced attention mechanism called Content Anchor Attention, which strengthens central features and captures contextual relationships between edge pixels, thereby enhancing the model’s classification and localization abilities.We augmented the YOLOv8 detection head with Dynamic Head to improve the detection head’s scale awareness, spatial awareness, and task awareness for small objects in images. This enhancement increased the model’s ability to perceive small objects, optimizing the detector’s performance.During model training, we introduced the Slideloss classification loss function and the ShapeIoU localization loss function. These functions strengthen the model’s learning of difficult samples, address issues arising from sensitivity of IoU to small object shifts, and accelerate the model’s convergence.

The rest of this paper is organized as follows. Section 2 presents some work related to this study. Section 3 presents the framework of the deep learning network used and the methods proposed in this work. In Section 4, the experimental results are provided and analyzed. Finally, the paper is concluded in Section 5.

## 2. Related Work

### 2.1. Yolo Algorithm

Since its release, the YOLO object detector has been renowned for its fast detection speed and has achieved an excellent balance between detection speed and accuracy through successive improvements. Redmon et al. introduced YOLOv1 [9] with the feature extraction network Darknet24 in 2016. YOLOv2 [20] later introduced the anchor boxes mechanism and selected a deeper backbone network for feature extraction. YOLOv3 [21] adopted multi-scale prediction, enabling the detection of objects at different scales. YOLOv4 [22] improved detection performance by using the advanced CSPDarknet. YOLOv5, while maintaining model accuracy, focused on optimizing inference speed, making it ideal for real-time applications. YOLOv6 [23] replaced the SPPF in the backbone with SimSPPF, which better extracts global features. The E-ELAN module in YOLOv7 [24] accelerates model convergence and reduces training time. YOLOv8 removed the anchor box mechanism and adopted the TOOD [25] strategy for positive and negative sample allocation, further optimizing the model. YOLOv9 [26] effectively addresses the problem of information loss in deep neural networks. YOLOv10 [27] boldly eliminates the NMS post-processing operation and directly produces detection results, becoming the first end-to-end detector in the YOLO series. The recently released YOLO11 adopts an improved backbone and neck architecture, enhancing feature extraction capabilities to achieve more accurate object detection and complex task performance. With continuous model improvements, the YOLO series has consistently optimized both accuracy and speed, leading the development of object detectors in the first stage.

### 2.2. Multi-Scale Feature Fusion

Multi-scale feature fusion can capture the details and characteristics of target objects at different scales, providing varied contextual information from different scales. By integrating information from multiple scales, it can effectively consolidate contextual information and improve the performance of recognition tasks. In computer vision tasks, multi-scale feature fusion can better handle scale variations and provide more accurate location and positional information. As a result, building multi-scale feature representations has consistently been a central research area in object detection. Since the advent of the feature pyramid network (FPN) [28], it has provided an effective solution for multi-scale feature fusion. This architecture enhances the feature extraction capabilities of CNN networks, ensuring that the final output features more accurately capture the dimensional information of the input image. The basic process involves three steps: the bottom-up pathway, which generates features of different dimensions from the bottom up; the top-down pathway, which enhances features from top to bottom; and the lateral connections between the CNN network layers and the final output features, using 1 × 1 convolutions to generate optimal output features. Building on the influence of FPN, PANet [29] adds a downsampling pathway to FPN, incorporating more semantic information into shallow features. NAS-FPN [30] utilizes neural architecture search to discover a new feature pyramid structure within a novel, scalable space that encompasses all cross-scale connections. However, finding the optimal architecture involves significant search costs. The authors of BiFPN [31] observed that the contributions of features at different scales to the fused output are often uneven. To address this, they proposed BiFPN, a simple yet effective weighted bidirectional feature pyramid network that introduces learnable weights to assess the significance of various input features and repeatedly applies top-down and bottom-up multi-scale feature fusion. In contrast, some researchers argue that the simple stacking of BiFPN modules is not optimal, as each BiFPN module operates independently, and deep stacking may risk gradient vanishing. Based on this, GFPN [32] was proposed, designing a new mechanism for skip-layer connection and cross-scale connection to balance accuracy and efficiency during model expansion. Another approach, AFPN [33], introduced a progressive feature pyramid network to support direct interactions between non-adjacent levels, avoiding significant semantic gaps between non-adjacent levels. Similarly, to ensure that the neck can transmit information losslessly during cross-layer fusion, Goldyolo [34] proposed a new collection and distribution mechanism composed of a Feature Alignment Module (FAM), an Information Fusion Module (IFM), and an Information Injection Module (Inject). This mechanism globally fuses multi-scale features and injects global attention into the higher levels, significantly enhancing the neck’s information fusion capability and improving model performance across different object sizes.

### 2.3. Attention Mechanism

The concept of attention mechanisms originated in the 1990s when cognitive scientists discovered that humans naturally filter out less relevant information and focus on information of interest. This process was termed the “attention mechanism”. In traditional machine learning, attention is exemplified through feature engineering, where input data are transformed into numerical vectors, helping the model select effective, appropriately scaled features. In 2014, Volodymyr Mnih introduced attention mechanisms to the vision domain with the paper “Recurrent Models of Visual Attention” [35]. This was followed by Ashish Vaswani’s 2017 paper “Attention is All You Need” [36], which proposed the Transformer structure. Since then, attention mechanisms have been widely applied in network designs for NLP and CV-related problems. Hu et al. proposed SENet [37] (Squeeze and Excitation Network), which applies attention to the channels of feature maps, enabling the model to learn the weights between different channels. This approach is simple, convenient, and adds only a small amount of computational and parameter overhead. GSoPNet [38] improved the squeeze part of SENet by replacing the original first-order average pooling with second-order pooling, enhancing global statistical modeling compared to SENet. The CBAM [39] attention mechanism is a hybrid attention mechanism that first processes the input feature map through a channel attention module and then through a spatial attention module. This improves the model’s perception ability and performance without increasing network complexity. However, CBAM’s approach to leveraging positional information by reducing the number of channels and using large-size convolutions primarily captures local correlations, failing to model long-range dependencies, which are essential for visual tasks. Coordinate Attention (CA) [40] encodes channel relationships and long-range dependencies by incorporating precise positional information. It achieves this by decomposing global pooling into the height and width directions, capturing cross-channel information as well as direction-aware and position-aware data, which helps the model more accurately locate and identify targets of interest. Triplet Attention [41] includes three parallel branches: two focus on capturing cross-dimensional interactions between channels (C) and spatial dimensions (H or W), while the third branch constructs spatial attention. The outputs of these three branches are averaged to mitigate the separation issue between channel attention and spatial attention. SimAM [42] leverages the property that adjacent pixels in an image have high similarity, while distant pixels have low similarity. It generates attention weights by calculating the similarity between each pixel in the feature map and its neighboring pixels.

### 2.4. Detection Head

The detection head is the final component of an object detection network, responsible for generating the final detection results, including object categories and locations. The detection head enables features extracted by the backbone network, initially designed for image classification, to be effectively transformed into features suitable for object detection. This further enhances the network’s performance in handling complex scenes and diverse objects, making it a key factor in improving detection accuracy. Early object detectors, such as YOLOv3 and YOLOv4, used a coupled head, where the detection head simultaneously predicted both object categories and bounding box locations. However, this design has some limitations. Song et al. [43] suggested that category prediction and location detection focus on different regions of interest in the image. Category prediction is a classification problem that focuses on the object’s center, while location detection is a regression problem that focuses on the object’s edges. Moreover, combining category and location detection into a single head may cause errors in one task to affect the other. To address this, Wu et al. [44] introduced a decoupled head, employing two separate branches within the detection head: one for predicting categories and the other for predicting locations. The subsequent YOLOX [45] model also adopted this decoupled head, achieving notable improvements in detection performance. Zhuang et al. [46] propose a novel Task-Specific Context Decoupling head. They input feature maps rich in semantic information into the classification branch and high-resolution feature maps containing more edge information into the localization branch to enhance detection performance. Jiang et al. [47] argued that the classification and localization branches in typical decoupled heads lack sufficient interaction. To address this, they proposed a new detection head, G-Head, which enhances the interaction between different tasks and facilitates the process of multi-task learning.

## 3. Method

YOLOv8 is one of the most advanced object detection algorithms available today, employing state-of-the-art methods in positive and negative sample selection strategies, feature extraction, and multi-scale feature fusion, making it suitable for most everyday scenarios. However, due to the characteristics of UAV perspective images, such as the prevalence of small targets, dense and uneven distribution, YOLOv8 struggles to achieve satisfactory detection performance in these images, rendering it inadequate for UAV aerial scenes. To address these issues, we have optimized the YOLOv8s base model by focusing on multi-scale feature fusion, detection heads, attention mechanisms, and loss functions. Our main improvement strategies are as follows: First, we introduced the concept of the High-level Screening-feature Pyramid Network (HS-FPN) [48] to enhance the neck of YOLOv8 that integrates multi-scale features, and then upgraded it to HS-PAN, similar to PANet, to better extract semantic information from input images. Additionally, we replaced the original channel attention mechanism with Context Anchor Attention [49], which is more suitable for small targets, thereby making the model more focused on small object features. Furthermore, we integrated an additional detection head after the 160 × 160 feature map in the original YOLOv8, and replaced all detection heads with Dynamic Head [50], which possesses scale awareness, spatial awareness, and task awareness, thus enhancing the detection heads’ perception of small objects. Finally, we improved the classification and localization loss functions during model training to ensure the better convergence of the model. The overall structure of our proposed model is shown in Figure 1.

### 3.1. CAA-HSPAN (Context Anchor Attention High-Level Screening-Feature Path Aggregation Network)

In UAV-perspective images, the objects to be detected exist at different scales. Even objects of the same category can vary in size and features due to distance variations within the same viewpoint, presenting a significant challenge for successful detection. To address the multi-scale issue inherent in UAV perspective images, we designed the Context Anchor Attention High-level Screening-feature Path Aggregation Network (CAA-PAN) to achieve multi-scale feature fusion, as detailed in Figure 2. This network structure is an improvement based on HS-FPN, and experimental results demonstrate that this design enables the model to capture features in UAV images more comprehensively, thereby improving the model’s performance.

#### 3.1.1. Feature Enhancement Module

The feature selection module of HS-FPN consists of the CA module and Dimensional Matching (DM). The CA module focuses on extracting the most representative information from each channel. However, we argue that extracting valuable spatial information without adding extra parameters is more advantageous for object detection in UAV-perspective images. Therefore, we replaced the original Channel Attention (CA) module with the Context Anchor Attention (CAA) module. The CAA module enhances central features while grasping contextual interdependencies among distant pixels, which is highly beneficial for small object localization. The CAA module first performs an average pooling and a 1 × 1 convolution operation on the input feature map to obtain local regional features:(1)$$F^{pool}=Conv_{1\times 1}\left(P_{avg}\left(X\right)\right),$$
where $P_{avg}$ denotes average pooling and X represents the input feature map. Then, two depthwise strip convolutions are used to replace the standard depthwise convolution.(2)$$F^{w}=DWConv_{1\times k_{b}}\left(F^{pool}\right),$$(3)$$F^{w}=DWConv_{1\times k_{b}}\left(F^{pool}\right),$$

There are three main reasons for using depthwise strip convolutions. First, strip convolutions are a lightweight convolution method. Compared to standard 2D depthwise convolutions, using two 1D strip convolutions can significantly reduce computational costs while achieving similar effects. Second, strip convolutions enhance the recognition and extraction of elongated target features, such as cars and pedestrians. Third, strip convolutions can connect distant pixels without significantly increasing computational complexity.

Finally, the attention weights are created using a Sigmoid function and then multiplied with the input feature map.(4)$$A=Sigmoid\left(Conv_{1\times 1}\left(F^{h}\right)\right),$$(5)$$F^{att}=A\times X,$$

Subsequently, feature fusion is performed, but first, dimensional matching is needed for the feature maps of different scales due to their varying numbers of feature channels. Here, the DM module uses 1 × 1 convolutions to adjust the number of channels for each scale’s feature map to 256.

#### 3.1.2. Feature Fusion Module

Features from the backbone network typically exhibit two key traits: high-level features are rich in semantic information but lack precise target location details, while low-level features provide accurate location information but are less effective in semantic classification. To address these issues, a common approach is to use a feature pyramid structure, upsample the high-level feature maps, and then add or concatenate them with low-level feature maps. Subsequently, the resulting feature maps are downsampled and again added or concatenated with the previous feature maps. However, this approach lacks a feature selection phase, simply performing elementwise addition or channel concatenation directly. Therefore, we designed the USFF and DSSF modules. The USFF module employs feature maps with rich semantic information as weights to filter out irrelevant semantic details from the low-level feature maps. Conversely, the DSSF module uses feature maps containing spatial location information as weights to improve the spatial accuracy of high-level features. As shown in Figure 3a, given a high-level $f_{high}\in R^{C\times H\times W}$ feature map and a low-level $f_{low}\in R^{C\times 2H\times 2W}$ feature map, the high-level feature map is first subjected to a transpose convolution operation with a stride of 2, doubling the height and width of the feature map to $f_{high^{\prime }}\in R^{C\times 2H\times 2W}$.

Next, the CAA module is applied to transform the features into corresponding weights, which are then used to filter the low-level feature maps. Finally, the filtered feature maps are fused with the high-level features $f_{out}\in R^{C\times 2H\times 2W}$, enhancing the model’s feature representation capability; the DSSF module in Figure 3b operates similarly to the USFF module, with the key difference being that larger feature maps are downsampled before being processed by the CAA module to filter the smaller feature maps. The process of feature selection and fusion is as follows:


(6)
$$f_{1}=TConv\left(f\right),$$



(7)
$$f_{out}=CAA\left(f_{1}\right)\times f_{2}+f_{1},$$


### 3.2. Dynamic Head

To improve the network’s capability in identifying small-scale objects, we added a detection head after the p2 layer, a 4× downsampled feature map. This is because, when downsampling small objects by 4×, most of the features of the small objects have not completely disappeared from the feature map. The 4× downsampled feature map can well preserve the spatial and semantic features of small objects, which is very beneficial for UAV-perspective object detection tasks. In addition, we replaced the four detection heads with dynamic heads, effectively improving the performance of the object detector. The detailed structure of the Dyhead is shown in Figure 4.

For object detection in UAV perspectives, an image often contains multiple objects of different scales, so the head should be scale-aware. Additionally, since objects can have different shapes, angles, and positions in different views, the head should be spatial-aware. Moreover, because objects can be expressed in various forms, such as a bounding box, center, points, etc., each with different target functions and optimization methods, the head should be task-aware. The dynamic head can unify scale awareness, spatial awareness, and task awareness. Suppose the feature map output from the backbone is level*space*channel. The dynamic head uses different attention mechanisms across different dimensions: attention in the level dimension can learn the relative importance of semantic layers and enhance object features in UAV-perspective images based on scale; attention in the spatial dimension can learn distinctive feature representations in space; and attention in the channel dimension can improve the feature representation capability of different channels. For the multi-scale L-layer feature maps $F_{in}={\left\{F_{i}\right\}}_{i=1}^{L}$ output from the backbone, they are first resized to the size of the middle-scale feature map, resulting in a 4D tensor $F\in R^{L\times H\times W\times C}$, where L represents the number of feature maps input to the detection head, and H, W, C represent the height, width, and feature channels of the middle-scale feature map, respectively. Then, define S=H×W, and reshape the 4D feature into a 3D feature $F\in R^{L\times S\times C}$. Finally, apply self-attention to it, with the general formula as follows:(8)W(F)=π(F)·F,

π(·) is an attention function. Nonetheless, directly learning the attention function across all dimensions proves to be computationally demanding and necessitates considerable memory resources. Consequently, the process is divided into three sequential attentions, with each sub-attention dedicated to a single dimension:(9)$$W\left(F\right)=\pi_{C}\left(\pi_{S}\left(\pi_{L}\left(F\right)\cdot F\right)\cdot F\right)\cdot F,$$

In the formula, $\pi_{L}\left(\cdot \right)$, $\pi_{S}\left(\cdot \right),\;\pi_{C}\left(\cdot \right)$ are different attention mechanisms applied to the L, S, and C dimensions, respectively.

#### 3.2.1. Scale-Aware Attention

Utilize scale-aware attention to adaptively integrate features of varying scales according to their semantic significance.(10)$$\pi_{L}\left(F\right)\cdot F=\sigma \left(f\left(\frac{1}{SC}\sum_{S,C}F\right)\right)\cdot F,$$

f(·) is a linear function implemented using a 1 × 1 convolution, and $\sigma \left(x\right)=\max\left(0,\min\left(1,\frac{x+1}{2}\right)\right)$ is a hard sigmoid function.

#### 3.2.2. Spatial-Aware Attention

Using spatial-aware attention, the network is guided to focus on abrupt changes in spatial positions and feature levels. Given the large S dimension, this involves two steps: first, using deformable convolutions to enable the attention to learn the coefficient features, and subsequently aggregating features at corresponding locations across different scale feature maps.(11)$$\pi_{S}\left(F\right)\cdot F=\frac{1}{L}\sum_{l=1}^{L}\sum_{k=1}^{K}\omega_{l,k}\cdot F\left(l;p_{k}+\Delta p_{k};c\right)\cdot \Delta m_{k},$$

Here, K is the number of sparse sampling locations, $p_{k}+\Delta p_{k}$ is the sampling position after learning the offset, allowing the attention mechanism to focus on abrupt changes, and $\Delta m_{k}$ represents the importance learned at position $p_{k}$.

#### 3.2.3. Task-Aware Attention

By employing task-aware attention, we can dynamically control each channel for different tasks, using the most appropriate channels for predictions.(12)$$\pi_{C}\left(F\right)\cdot F=\max\left(\alpha^{1}\left(F\right)\cdot F_{C}+\beta^{1}\left(F\right),\alpha^{2}\left(F\right)\cdot F_{C}+\beta^{2}\left(F\right)\right),$$

$F_{C}$ is the slice on the C-th channel, $\left[\alpha^{1},\alpha^{2},\beta^{1},\beta^{2}\right]=\theta \left(\cdot \right)$ are hyperparameters that learn to control the activation threshold; initially, global average pooling is applied across the L*S dimensions to reduce dimensionality. This is followed by two fully connected layers, a normalization layer, and a sigmoid activation function, normalizing to [−1, 1].

### 3.3. Loss Function

In aerial drone scenarios for object detection tasks, small objects comprise a significant proportion. An appropriately designed loss function can substantially improve the model’s detection performance. YOLOv8 employs DFL [51] and CIoU [52] losses for bounding box regression, alongside BCE loss for classification. However, BCE loss does not effectively balance between challenging and simpler samples. To address this, we introduce SlideLoss [53], which differentiates between hard and easy samples based on the Intersection over Union (IoU) [18] between predicted and ground truth boxes. To reduce hyperparameters, the loss uses the average IoU of all bounding boxes as a threshold µ; samples with classification scores below µ are considered hard, and those above µ are considered easy. Moreover, ambiguous samples near classification boundaries often incur large losses, which SlideLoss guides the model to optimize and effectively utilize for training. Given their scarcity, these samples receive greater weight during loss computation. The specific process involves determining hard and easy samples using parameter µ, followed by a weighting function that emphasizes samples with ambiguous classifications.(13)$$f\left(x\right)=\{\begin{array}{c} 1,\;\;\;\;\;\;\;\;x\leq \mu -0.1\\ e^{1-\mu },\;\mu <x<\mu -0.1\\ e^{1-x},\;\;\;\;\;\;\;\;\;\;x\geq \mu \end{array},$$

In object detection, bounding box regression losses are typically measured using IoU and its various variants, including GIoU [54], DIoU [55], CIoU, EIoU [56], and SIoU [57], among others. Although these methods account for factors such as the distance, shape, and angle between predicted and ground truth boxes, they often neglect the effects of the bounding box’s shape and scale itself. In Figure 5a, although the deviation between A and B is identical to that between C and D, the Intersection over Union (IoU) values differ. Similarly, while the deviation between C and D remains consistent, the IoU values vary. However, the difference in IoU values between C and D is less pronounced compared to the difference between A and B. In Figure 5b, a similar pattern is observed between A and B, as well as between C and D. In contrast, Figure 5a illustrates that the variation in IoU between A and B is attributable to differences in the shapes of the ground truth boxes, with deviations corresponding, respectively, to the directions of the long and short sides. It can be seen that deviations along the direction of the long side have a smaller impact on IoU compared to deviations along the direction of the short side. In Figure 5b, it is observed that the shape of the ground truth affects the variation in IoU values during regression. Here, we use Shapeiou [58] to address the above-mentioned issues. Its computational formula is as follows:(14)$$IoU=\frac{\left|B\cap B^{gt}\right|}{\left|B\cup B^{gt}\right|},$$(15)$$ww=\frac{2\times {\left(w^{gt}\right)}^{scale}}{{\left(w^{gt}\right)}^{scale}+{\left(h^{gt}\right)}^{scale}},$$(16)$$hh=\frac{2\times {\left(h^{gt}\right)}^{scale}}{{\left(w^{gt}\right)}^{scale}+{\left(h^{gt}\right)}^{scale}},$$(17)$$distance^{shape}=hh\times {\left(x_{c}-x_{c}^{gt}\right)}^{2}/c^{2}+ww\times {\left(y_{c}-y_{c}^{gt}\right)}^{2}/c^{2},$$(18)$$\Omega^{shape}=\underset{t=w,h}{\sum }{\left(1-e^{-\omega_{t}}\right)}^{\theta },\theta =4,$$(19)$$\{\begin{array}{c} \omega_{w}=hh\times \frac{\left|w-w^{gt}\right|}{max\left(w,w^{gt}\right)}\\ \omega_{h}=ww\times \frac{\left|h-h^{gt}\right|}{max\left(h,h^{gt}\right)} \end{array},$$
where scale is a parameter related to the dataset, and ww and hh are the weight coefficients in the horizontal and vertical directions, respectively, which are related to the shape of the ground truth. The bounding box regression loss is as follows:(20)$$L_{Shape-IoU}=1-IoU+distance^{shape}+0.5\times \Omega^{shape},$$

Ultimately, we use SlideLoss and ShapeIoU+DFL loss as the classification loss and localization loss for our model, respectively. Experimental results demonstrate that this approach can improve the performance of the object detector.

## 4. Experiments

This chapter primarily introduces the dataset used in this study, the experimental environment, and the training strategies. Additionally, it presents the evaluation metrics related to the experimental results.

### 4.1. Datasets

The VisDrone2019 dataset [59], developed by Tianjin University and the AISKYEYE data mining team, is a prominent resource for aerial drone imagery. It encompasses scenes from over ten cities across China, captured using various drones under diverse angles, scenarios, and tasks. This results in a dataset rich in information, covering different environments (urban and rural), object types (pedestrians, vehicles, bicycles, etc.), and densities (sparse to crowded). The dataset includes 10,209 static images (6471 for training, 548 for validation, and 3190 for testing), with more than 2.6 million manually annotated bounding boxes. Additionally, it provides crucial attributes such as scene visibility, object categories, and occlusion conditions to enhance data utilization. The VisDrone2019 dataset is a high-quality dataset specifically designed for object detection from UAV perspectives. It effectively represents the scenarios that UAVs typically encounter in daily use, aligning with the content and issues addressed in this study. Therefore, we conducted comparative and ablation experiments on this dataset.

The AI-TODv1.5 dataset [60] is derived from several publicly available large-scale aerial image datasets, including the DOTA-v1.5, xView, VisDrone2018-Det, Airbus Ship, and DIOR datasets. The dataset is created by dividing the original images into 800 image patches with a 200-pixel overlap. Images smaller than 800 pixels are padded with zeros. This procedure results in 11,214 training images and 2804 validation images. The dataset encompasses eight categories: airplane, bridge, oil tank, ship, swimming pool, vehicle, person, and windmill. The average size of the objects in AI-TODv1.5 is just 12.7 pixels, presenting a substantial challenge for object detection. Compared to the VisDrone2019 dataset, the AI-TODv1.5 dataset contains objects with smaller sizes and requires the detection of objects on the ground or on water surfaces from higher altitudes, making it more challenging. Therefore, to further assess the effectiveness of our proposed model, we conducted ablation experiments using this dataset.

To evaluate the generalization performance of the proposed model, we also used the PASCAL VOC dataset to assess our model’s performance and compare it with other object detection algorithms. Compared to the two small object datasets mentioned above, the Pascal VOC dataset is a conventional object detection dataset, primarily consisting of ground-level images and featuring a wider range of detection categories. Experiments on this dataset can assess the generalization performance of the proposed model. This dataset combines the PASCAL VOC 2007 and 2012 sets and is specifically designed for object detection tasks, containing 20 object categories. For our experiments, we used 16,551 images from the VOC 2007 and VOC 2012 datasets for training and 4952 images from the VOC 2007 dataset for testing.

### 4.2. Experimental Environment and Training Strategies

The hardware and environment parameters we used during the model training phase are shown in the following Table 1.

We applied consistent hyperparameters across the training, validation, and testing phases of our model. The chosen optimizer was SGD, configured with a momentum of 0.937 and a weight decay of 5 × 10^−4^. We implemented a warm-up strategy during the initial three epochs of training. For the datasets mentioned, the learning rate decay factor (gamma) was set to 0.01.

To evaluate the detection performance of our proposed improved model, we used several metrics including precision, recall, mAP0.5, mAP0.5:0.95, the number of model parameters, model size, and detection speed. The following terms were utilized in the formulas for some of these evaluation metrics: TP (true positive, correctly predicted as positive), FP (false positive, incorrectly predicted as positive), and FN (false negative, incorrectly predicted as negative). Intersection over Union (IoU) measures the ratio of the overlap area to the union area of the predicted bounding box and the ground truth box. Precision is defined as the ratio of true positive predictions to the total number of detected samples, and it is calculated using the following formula:(21)$$Precision=\frac{TP}{TP+FP},$$

Recall is the ratio of the number of correctly predicted positive samples to the actual number of positive samples, calculated as shown in the following formula:(22)$$Recall=\frac{TP}{TP+FN},$$

Average precision (AP) is the area under the precision–recall curve, calculated as shown in the following formula:(23)$$AP=\int_{0}^{1}Precision\left(Recall\right)d\left(Recall\right),$$

Average precision (mAP) is the weighted average of AP values across all sample categories, used to measure the detection performance of a model across all classes. The formula is as follows:(24)$$mAP=\frac{1}{N}\sum_{i=1}^{N}AP_{i},$$

In Formula (24), $AP_{i}$ represents the AP value for the category index i, where N denotes the number of categories in the training dataset (in this context, N is 10). mAP0.5 indicates the average precision when the IoU threshold for the detection model is set to 0.5, while mAP0.5:0.95 denotes the average precision when the IoU threshold ranges from 0.5 to 0.95 (in increments of 0.5). GFLOPs measures the computational complexity during model training, representing billions of floating-point operations per second. Params refers to the number of parameters in the model, indicating the consumption of computational memory resources. FPS measures the number of images the model can detect per second, directly related to the resolution of the input images during detection. Generally, higher input image resolutions result in lower FPS under the same configuration.

### 4.3. Experimental Results on Visdrone2019 Dataset

To validate the effectiveness of our improved model, we performed comparative experiments with several state-of-the-art object detection algorithms using the VisDrone2019 dataset’s test set. The experiments used an input image resolution of 640 × 640, and the models were trained for 300 epochs. The comparative results of different models on the VisDrone2019 dataset are presented in Table 2.

Table 2 shows the detection results of our improved method compared to other object detection models. Our model utilizes CSPDarknet53 as the backbone network, CAAHSPAN as the neck, and Dyhead as the detection head, with an input image resolution of 640 × 640. Ultimately, it achieved an mAP50 of 41.4%, attaining the highest accuracy across each detected category. Notably, there was a significant improvement in detecting small objects, such as bicycles. Additionally, our model is an improvement based on the YOLOv8s model. As shown in the Table 2, the improved model not only surpasses the YOLOv8s model but also outperforms theYOLOv8m model while having fewer parameters and lower computational complexity (our proposed model has 10.1 million parameters and 72.7 GFLOPs, compared to YOLOv8m’s 25.9 million parameters and 79.1 GFLOPs).

We plotted the precision–recall (P-R) curves for each category, as illustrated in Figure 6. The AP for each class, listed in the table, corresponds to the area under the respective P-R curve. Compared to YOLOv8s and YOLOv8m, the proposed model achieved the highest mAP50, with the most significant improvements in the mAP for the “people” and “motor” categories. In Figure 7, we also present the confusion matrix based on our model’s results on the VisDrone2019 dataset. The prediction was conducted with an Intersection over Union (IoU) threshold of 0.5 and a confidence threshold of 0.25. The confusion matrix displays the classification outcomes for each category, with rows representing the model’s predictions and columns indicating the actual categories. The values along the diagonal indicate the proportion of correct classifications made by the model. The higher the probability on the diagonal, the better the model predicts that category. From Figure 7, it can be seen that most people were misclassified as background. This is due to the dense distribution of targets in this dataset, with most people crowded together, resulting in significant occlusion and small target sizes. When the model downsamples to extract features, it might misclassify these targets as background noise. Additionally, most bicycles, tricycles, and auto rickshaws were not correctly predicted, resulting in low AP values for these classes. Firstly, this is because the number of training samples for these classes is relatively small compared to other categories, limiting the model’s learning of these categories’ information. Secondly, these are small objects, making it challenging to extract useful information in environments with occlusion and dense targets, thus hindering the detection of these objects.

### 4.4. Experimental Results on the PASCAL VOC Dataset

To further assess the effectiveness of various networks, we carried out comparative experiments using the PASCAL VOC dataset. We used the training and validation images from PASCAL VOC 2012 along with the training images from VOC2007 as the training set, and validated the results on the VOC2007 test set.

Table 3 presents the comparative results of our proposed method and other popular object detectors. As can be seen from the table, our proposed model significantly outperforms two-stage object detection algorithms, such as Faster R-CNN. Additionally, among single-stage object detection algorithms, our model achieves the highest mAP50. Our model is an improved version of YOLOv8s, and it shows a slight reduction in the number of parameters, decreasing by nearly 1 million. Compared to YOLOv8s’s mAP50 of 81.5, our improved algorithm reaches an mAP50 of 82.4, an increase of nearly one point. Furthermore, in comparisons with the newly introduced YOLOv9s and YOLOv10s, our proposed algorithm still outperforms them, particularly YOLOv10s, where our method shows an improvement of nearly two points. Compared to YOLOv4 and YOLOv8m, these two models have significantly more parameters than the model we proposed. Additionally, our model’s advantages were not fully realized on conventional datasets, resulting in slightly lower performance metrics compared to these two models. Moreover although there is a decrease in FPS (frames per second), the 52.2 FPS achieved by our proposed algorithm is sufficient to meet the needs of daily life and production work.

### 4.5. Ablation Experiments

In this section, we conducted experiments on all modules of the model using the VisDrone2019 validation set to evaluate their impact on model performance, followed by a combination and step-by-step analysis of each module. Additionally, the loss function was evaluated. The ablation studies used the YOLOv8s algorithm as a baseline, with mAP, model size, parameter count, and computational complexity as evaluation metrics.

#### 4.5.1. Ablation Experiments on Visdrone2019 Val Dataest

In our experiments, we made slight improvements to the positive and negative sample assignment strategy of YOLOv8. YOLOv8 uses the TaskAlignedAssigner matching strategy, which selects positive samples based on the weighted scores of classification and regression. The formula is as follows:(25)$$align-metric=s^{\alpha }*u^{\beta },$$

Here, s is the predicted class score, u is the CIoU between the predicted box and the ground truth box, α and β are weight hyperparameters. The product of these 
terms measures the matching degree, with the align-metric approaching 1 
indicating a higher match between the predicted box and the ground truth box. 
For each ground truth box, we sort the align-metrics and select the top *k* predicted boxes as 
positive samples. By default, k is set to 10, but for the VisDrone2019 dataset, we modified k to 5. This adjustment was made because the targets in this dataset are densely packed, leading to significant overlap among the ground truth boxes. If k is set to 10, these 10 predicted boxes would match one ground truth box. However, due to the dense distribution of the ground truth boxes, not all 10 predicted boxes would appropriately correspond to the specific ground truth box; they might be better suited for the surrounding overlapping ground truth boxes. By reducing k to 5, the selected predicted boxes are more likely to correspond to the intended ground truth box. Experiments showed that setting k to 5 improved the model’s performance.

In the experiment, the input image resolution for the model was set to 640 × 640. The ablation test results are presented in Table 4. As shown in the table, when the topk value in TaskAlignedAssigner was adjusted from 10 to 5, all metrics saw slight improvements without altering the number of parameters or computations. Consequently, for experiments on the VisDrone2019 dataset, k was set to 5.

Additionally, we added a detection head after the 160 × 160 feature map. This layer of the feature map can better retain the spatial location information and semantic information of small targets, making it very suitable for small object detection. Despite a slight increase in computational complexity, the performance improvement is substantial, with mAP50 increasing from 40.9% to 45.5% and mAP50:95 rising from 24.4% to 27.7%. This indicates that for object detection tasks from a drone’s perspective, adding an extra detection head is both necessary and effective. Therefore, all subsequent experiments will be conducted with this additional detection head.

Subsequently, we replaced the neck part of the network with the CAA-HSPAN structure. Compared to the original PAN structure, mAP50 increased from 45.5% to 48.7%, and mAP50:95 rose from 27.7% to 29.9%, with both precision and recall improving by 2 to 3 points. This demonstrates that the CAA-HSPAN multi-scale feature fusion module, which integrates the CAA mechanism with HSPAN, can adaptively fuse information between deep and shallow feature maps and integrate contextual information, thus better extracting useful information. When fusing two feature maps, the CAA module is used to help the model focus more on the target regions in the input image, resulting in some computational overhead. Although the number of parameters decreased by 1 million, the computational complexity increased significantly, adding nearly 30 GFLOPs. However, this is still less than the computational complexity of YOLOv8m. The real-time performance of the model was also affected, with a decrease in FPS.

Next, we replaced the classification and localization losses in YOLOv8 with Slide Loss and ShapeIoU, respectively. This change improved *mAP*50 from 48.7% to 49.6%, with a 2-point increase in precision, without adding computational complexity or parameter count. This indicates that by changing the loss functions, the model pays more attention to hard-to-distinguish samples and effectively utilizes them during training, allowing the model to perform better in both classification and localization, thereby enhancing the performance of the detector.

Finally, we replaced all detection heads in our proposed model with Dyhead. This resulted in a slight increase in both the parameter count and computational complexity, with the parameter count increasing by 0.6 M and the computational complexity rising from 69.4 GFLOPs to 72.7 GFLOPs. However, this is still less than the computational complexity of YOLOv8m. After using Dyhead, mAP50 increased from 49.4% to 52.2%, and mAP50:95 improved from 30.3% to 31.8%, with the recall rate also increasing by nearly 2 points. This indicates that Dyhead can capture feature information at different scales, enabling the model to classify and localize targets accurately and efficiently. It improves the performance of object detection tasks from a drone’s perspective while maintaining a relatively small computational load.

#### 4.5.2. Ablation Experiments on AI-TOD Val Dataset

To fully verify the effectiveness of each module proposed in this paper, we also conducted ablation studies on the AI-TOD dataset. The information and results for each model are shown in Table 5.

From the table, it can be seen that when we added a detection head after the P2 layer, the model’s performance improved significantly, with increases in precision, recall, and mAP. Specifically, mAP50 increased from 38.8% to 41.7%, although FPS decreased. Furthermore, each time we incorporated one of our proposed modules into the model, all evaluation metrics showed improvements. When all the improvements were integrated into the network, the model achieved the best performance, with the highest precision and an mAP50 increase to 45.2%. Although the final model experienced a decrease in FPS to 52.2, it remains sufficient to meet the demands of everyday production and work needs.

### 4.6. Visualization

To more intuitively demonstrate the effectiveness of our proposed method in everyday scenarios, we selected some representative images from the VisDrone2019 test set for detection and conducted a visualization analysis. The visualization results and comparisons are shown in Figure 8.

In Figure 8, we present some representative test results. Figure 8a shows the results obtained using the YOLOv8s model, while Figure 8b shows the results predicted by our improved model. We have enlarged some detection results in the images to facilitate a more intuitive comparison. The targets in the images in Figure 8 are very dense, with many distant targets that have indistinct features, and the problem of occluded targets is quite severe.

From the detection results in Figure 8b, it is evident that our proposed algorithm can accurately detect targets even in densely distributed scenarios. It also performs well in identifying targets that are distant, with indistinct features, by recognizing some relatively closer objects. In high-altitude images, ground-level pedestrians and vehicles are typically very small, occupying only a small portion of the pixels. Additionally, different types of targets may overlap. Our proposed model effectively detects these targets, alleviating this issue. Moreover, in shadowed or dimly lit scenes, the baseline model missed many targets, failing to detect most of them. In contrast, our model could still detect most of the targets that the baseline model missed in darker scenes. These results demonstrate that our algorithm has good robustness.

This paper proposes a novel spatial attention-based feature fusion mechanism that effectively integrates information across different scales, enhances contextual information, and maximizes the retention of small object features. Compared to conventional spatial attention mechanisms, our approach achieves these benefits while effectively reducing model parameters and computational complexity, thereby preserving model performance. Additionally, the introduction of the dynamic head allows for perception across channels, spatial dimensions, and tasks, enabling more precise prediction of object categories and positions, thereby enhancing overall model performance. Furthermore, the Slideloss classification loss and ShapeIoU regression loss used during training facilitate faster and better convergence of the model. They improve the accuracy of object classification and the precision of object localization.

## 5. Conclusions

From an aerial drone perspective, images often exhibit complex scenes with dense clusters of targets, including numerous small objects. This complexity typically results in lower accuracy for conventional object detection algorithms, which consequently fail to meet practical application requirements in drone scenarios. This paper introduces an improved object detection algorithm based on YOLOv8s to address these challenges effectively. Firstly, in the “neck” part of the model, we adopt HS-PAN, an enhancement of HS-FPN with an additional downsampling branch, and introduce the context-aware attention spatial attention mechanism. This mechanism captures important features within image regions, promotes feature fusion across different levels, enhances contextual information, and improves the model’s accuracy in identifying small targets. Secondly, in the detection head, we incorporate channel-aware attention, spatial-aware attention, and task-aware attention to strengthen foreground features, reduce background noise interference in the final detection results, and effectively enhance the model’s ability to extract key information from small targets. Finally, during model training, we utilize Slideloss for classification loss and ShapeIoU for regression loss. These choices enable the model to focus more on challenging samples during training and mitigate the impact of target shape and size on IoU during regression, thereby accelerating convergence. Experiments were conducted on the VisDrone2019, AI-TOD, and PASCAL VOC datasets to evaluate the effectiveness of the proposed model. The results demonstrate that our model outperforms the YOLOv8s baseline model, confirming its effectiveness in detecting small objects within drone imagery.

Typically, higher accuracy often implies more parameters and larger computational requirements, demanding substantial computing resources. Moreover, excessive parameters can adversely affect detection speed, making real-time detection impractical. Therefore, our future main research direction will focus on improving model performance and accelerating detection speed while ensuring computational or parameter constraints are maintained.

## Figures and Tables

**Figure 1 sensors-25-00589-f001:**
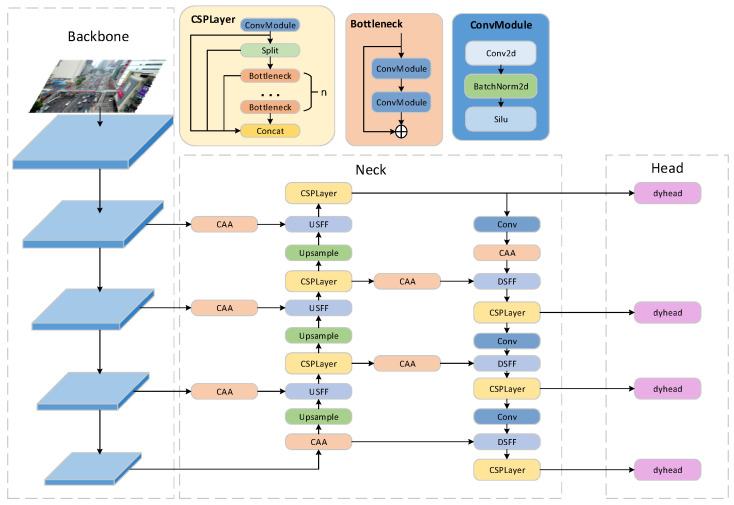
The architecture of our proposed model features the CSPDarknet53 network as the backbone for feature extraction. In the neck section, we incorporate a multi-scale feature fusion structure enhanced by CAA. In its head part, dynamic head is introduced.

**Figure 2 sensors-25-00589-f002:**
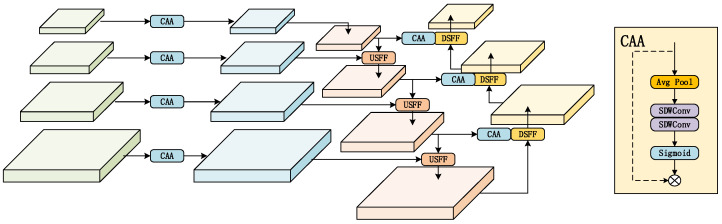
The structure of CAA-HSPAN. It consists of a feature fusion module and a feature enhancement module, where context anchor attention is a type of spatial attention. But in the USFF and DSFF modules, only attention weights are generated and not multiplied element by element with the input feature map.

**Figure 3 sensors-25-00589-f003:**
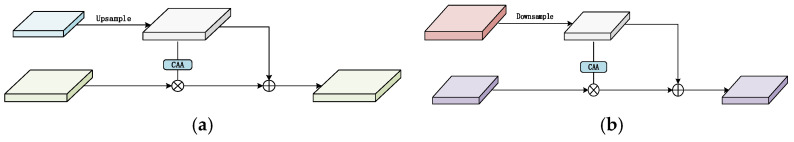
Two feature fusion models. (**a**) The structure of USFF; (**b**) The structure of DSFF.

**Figure 4 sensors-25-00589-f004:**
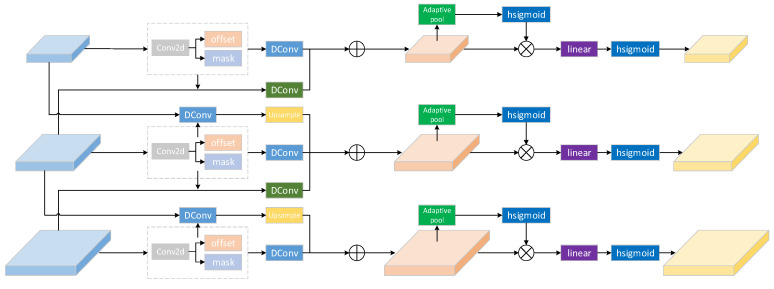
The overall structure of the dyhead (assuming 3 detection heads) consists of three parts: scale-aware attention, spatial-aware attention, and task-aware attention. The DConv filled in blue is a 3 × 3 deformable convolution with stride = 1, while the DConv filled in green is a 3 × 3 deformable convolution with stride = 2. Sigmoid refers to the hard sigmoid function.

**Figure 5 sensors-25-00589-f005:**
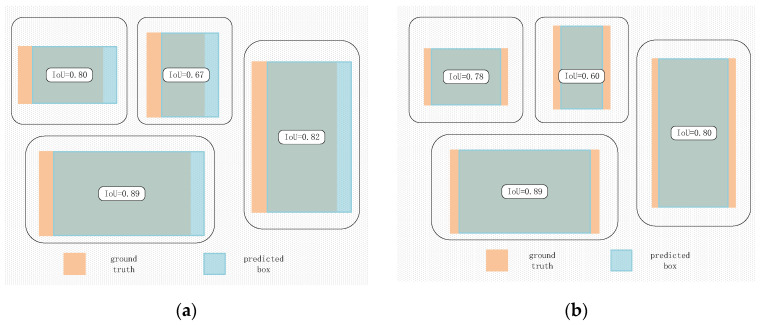
Analysis of Bounding Box Regression Characteristics. (**a**) The two bounding boxes only have coincidence relationship but no inclusion relationship; (**b**) The two bounding boxes not only have coincidence relationship but also have inclusion relationship.

**Figure 6 sensors-25-00589-f006:**
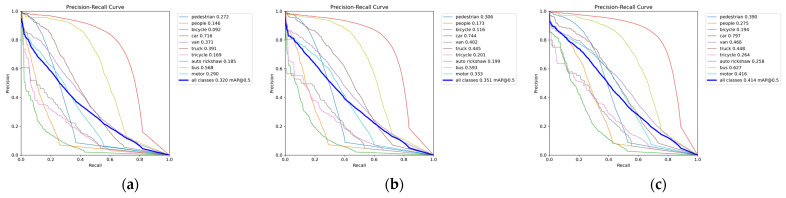
The P-R curve of our proposed model and baseline models on the Visdrone2019 dataset. (**a**) The P-R curve of YOLOv8s; (**b**) the P-R curve of YOLOv8m; (**c**) the P-R curve of our proposed model.

**Figure 7 sensors-25-00589-f007:**
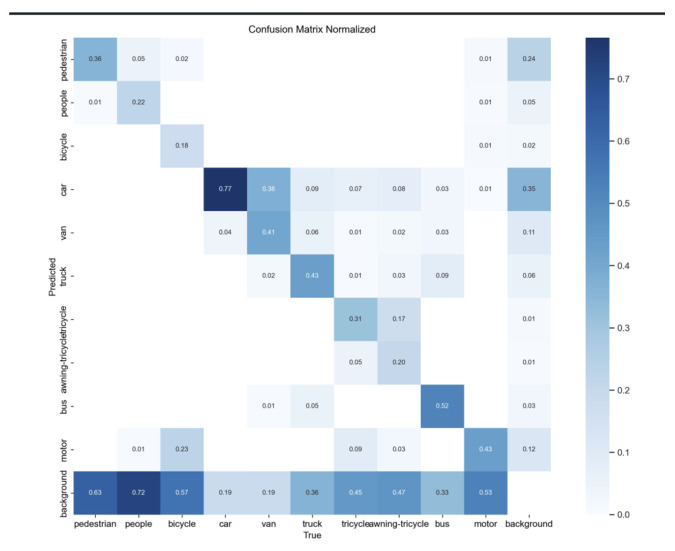
The P-R curve based on our model’s results on the VisDrone2019 test dataset, along with the confusion matrix for the same dataset are presented with an IoU threshold of 0.5 and a confidence thresh old of 0.25.

**Figure 8 sensors-25-00589-f008:**
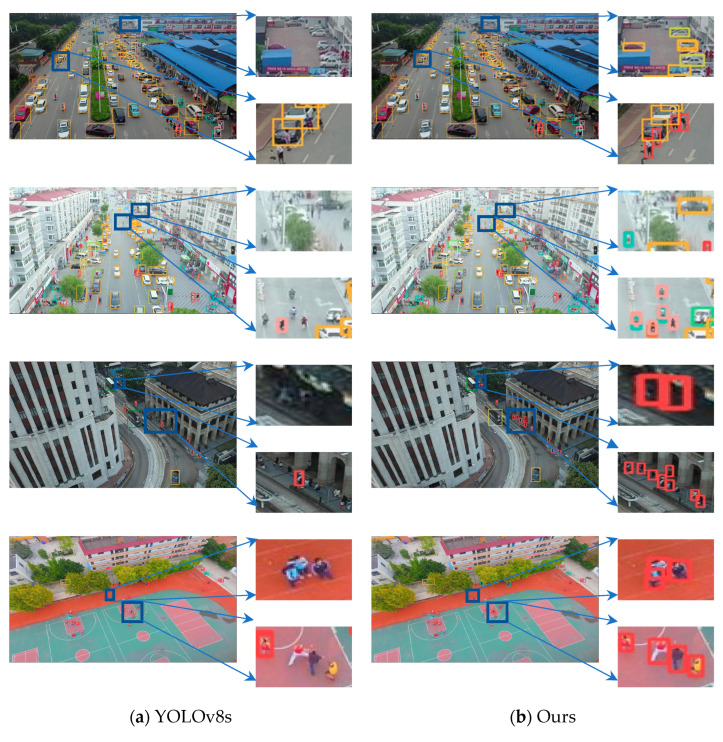
Visualization of results on images from the VisDrone2019 test dataset is presented. In these images, different categories are marked with boxes of different colors.

**Table 1 sensors-25-00589-t001:** Training environment and hardware platform parameters table.

Parameters	Configuration
CPU	I5-13600KF
GPU	NVIDIA GeForce RTX 3060ti
GPU memory size	8 G
Operating system	Win 11
Deep learning architecture	Pytorch2.0.0 + Cuda11.8

**Table 2 sensors-25-00589-t002:** Comparison of various methods on the Visdrone2019 test dataset.

Model	Object Category										mAP50 (%)
	PED	PER	BC	Car	Van	Truck	TRI	ATRI	Bus	MO	
Fast R-CNN [6]	21.4	15.6	6.7	51.7	29.5	19	13.1	7.7	31.4	20.7	21.7
Faster R-CNN [7]	20.9	14.8	7.3	51	29.7	19.5	14	8.8	30.5	21.2	21.8
CascadeR-CNN [61]	22.2	14.8	7.6	54.6	31.5	21.6	14.8	8.6	34.9	21.4	23.2
RetinaNet [61]	13	7.9	1.4	45.5	19.9	11.5	6.3	4.2	17.8	11.8	13.9
CenterNet [61]	22.6	20.6	14.6	59.7	24	21.3	20.1	17.4	37.9	23.7	26.2
DMNet [61]	28.5	20.4	15.9	56.8	37.9	30.1	22.6	14	47.1	29.2	30.3
HRDet+ [61]	28.6	14.5	11.7	49.4	37.1	35.2	28.8	21.9	43.3	23.5	28
ACM-OD [59]	30.8	15.5	10.3	52.7	38.9	33.2	26.9	21.9	41.4	24.9	29.7
CDNet [59]	35.6	19.2	13.8	55.8	42.1	38.2	33	25.4	49.5	29.3	34.2
HR-Cascade++ [61]	32.6	17.3	11.1	54.7	42.4	35.3	32.7	24.1	46.5	28.2	32.5
MSC-CenterNet [61]	33.7	15.2	12.1	55.2	40.5	34.1	29.2	21.6	42.2	27.5	31.1
YOLOv3-LITE [62]	34.5	23.4	7.9	70.8	31.3	21.9	15.3	6.2	40.9	32.7	28.5
MSA-YOLO [63]	33.4	17.3	11.2	76.8	41.5	41.4	14.8	18.4	60.9	31	34.7
YOLOv4 [21]	24.8	12.6	8.6	64.3	22.4	22.7	11.4	7.6	44.3	21.7	30.7
YOLOv5s	24.2	15.9	7.6	68.7	30.5	30.5	11.1	14.8	52.5	21.8	27.7
YOLOv8s	27.2	14.6	9.2	71.6	37.1	39.1	16.9	18.5	56.8	29	32
YOLOv8m	30.6	17.3	11.6	74.4	40.2	44.5	20.1	19.9	59.3	33.3	35.1
YOLOv9s [25]	27.5	15.3	9.2	71.4	37.6	39.6	19.1	20.4	58.2	30	32.8
YOLOv10s [26]	26.4	16.3	7.7	70.7	35.5	37.2	16.5	18	55.6	28.3	31.2
YOLO11s	27.6	13.9	8.55	71.7	37.3	39.7	18.4	19	57.4	29.3	32.2
Ours (test)	39.1	27.5	19.4	79.7	46.6	44.8	26.4	25.8	62.7	41.6	41.4

**Table 3 sensors-25-00589-t003:** Comparison of various methods on the VOC2007 test dataset.

Model	Backbone	Size	Parameters	mAP50	FPS
Fast R-CNN [6]	ResNet	600 × 1000	44.5 M	70.0	1.9
Faster R-CNN [7]	ResNet	600 × 1000	40.1 M	76.4	2.4
CascadeR-CNN [64]	ResNet	600 × 1000	76.8 M	79.6	2.2
CenterNet [65]	ResNet	512 × 512	12.7 M	77.1	7
RetinaNet [10]	ResNet	600 × 600	33.8 M	81.5	5
YOLO [61]	GoogleNet	448 × 448	62 M	63.4	45
YOLOv2 [61]	DarkNet	352 × 352	67 M	76.8	67
YOLOv3 [20]	DarkNet	416 × 416	61.5 M	78.3	36.9
YOLOv3-LITE [62]	DarkNet	416 × 416	31 M	74.0	71.9
YOLOv4 [21]	CSPDarkNet	640 × 640	39.6 M	83.2	60
MSA-YOLO [63]	CSPDarkNet	640 × 640	12.1	80.4	174.9
CDNet [66]	CSPDarkNet	640 × 640	21.5 M	80.3	165.3
YOLOv5s	CSPDarkNet	640 × 640	9.13 M	79.9	210.5
YOLOv8s	CSPDarkNet	640 × 640	11.2 M	81.5	200.7
YOLOv8m	CSPDarkNet	640 × 640	25.9 M	84.5	80.7
YOLOv9s [25]	GELAN	640 × 640	7.3 M	82.2	101.1
YOLOv10s [26]	CSPDarkNet	640 × 640	8.07 M	80.8	172.8
YOLO11s	CSPDarkNet	640 × 640	9.4 M	81.6	212.2
Ours	CSPDarkNet	640 × 640	10.1 M	82.4	52.2

All the algorithms were trained using the combined training set of VOC2007 and VOC2012 and evaluated on the VOC2007 test dataset. Ultimately, the detection accuracy and processing speed of these detectors were compared.

**Table 4 sensors-25-00589-t004:** The effect of combining different modules in YOLOv8s on the VisDrone2019 test dataset (p2 refers to the additional prediction head).

**Method**	**mAP50 (%)**	**mAP50:95 (%)**	**P (%)**	**R (%)**	**GFLOPs**	**Params (M)**	**FPS**
YOLOv8s	39.3	23.5	50.9	38.2	28.7	11.1	200
YOLOv8s-topk = 5	40.9	24.4	52.1	39.9	28.7	11.1	200
YOLOv8s-p2-topk = 5	45.5	27.7	55.9	43.7	37	10.6	146
YOLOv8s-p2-topk = 5-CAAHSPAN	48.7	29.9	57.2	46.9	69.4	9.53	75
YOLOv8s-p2-topk = 5-CAAHSPAN- Slideloss-ShapeIoU	49.4	30.3	59.6	47.1	69.4	9.53	75
YOLOv8s-p2-topk = 5-CAAHSPAN- Slideloss-ShapeIoU-DyHead	52.2	31.8	60.9	49.5	72.7	10.1	52

**Table 5 sensors-25-00589-t005:** The effect of combining different modules in YOLOv8s on the AI-TODv1.5 validation dataset (p2 refers to the additional prediction head).

Method	mAP50 (%)	mAP50:95 (%)	P (%)	R (%)	GFLOPs	Params (M)	FPS
YOLOv8s	38.8	16.2	58.1	37.7	28.7	11.1	200
YOLOv8s-p2-topk = 5	41.7	18.3	60.9	40.4	37	10.6	146
YOLOv8s-p2-topk = 5-CAAHSPAN	43.3	19.3	62.9	40.7	69.4	9.53	75
YOLOv8s-p2-topk = 5-CAAHSPAN-Slideloss-ShapeIoU	44.1	19.6	63.1	41.9	69.4	9.53	75
YOLOv8s-p2-topk = 5-CAAHSPAN-Slideloss-ShapeIoU-DyHead	45.2	20.1	63.9	43.7	72.7	10.1	52

## Data Availability

Data are contained within the article.

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
