# Peer review of "A Feature-Enhanced Small Object Detection Algorithm Based on Attention Mechanism"

_sensors, 2025, doi:10.3390/s25020589_

Round 1

Reviewer 1 Report

Comments and Suggestions for Authors

This paper designs innovative modules to address issues such as small, dense, and unevenly distributed targets in images captured from an unmanned aerial vehicle perspective , which exhibit good detection performance. But a few doubts require further clarification and explanation  as followed:

1: What are the reasons for initially choosing the YOLOv8 model. It’s suggested to briefly explain reasons such as its superiority.

2:What is the relationship between the AI-TODv1.5 dataset and the VisDrone2019 dataset? Why is the AI-TODv1.5 dataset used for ablation experiments? How does this ablation experiment validate the effectiveness of the model?

3:Section 4.4 conducts comparative tests based on the PascalVOC dataset to ensure greater reliability and accuracy. It is recommended that the models compared be consistent with those in Table 1.

Author Response

Comments 1: What are the reasons for initially choosing the YOLOv8 model. It’s suggested to briefly explain reasons such as its superiority.

Response 1: Thank you for pointing this out. YOLOv8 has undergone extensive application and validation, and has been used in many practical projects with good results. Its technology and framework are highly mature, offering a high level of stability and suitability for various real-world applications. Additionally, YOLOv8 is capable of delivering excellent inference speed on edge devices and in real-time scenarios while maintaining high accuracy.

Comments 2: What is the relationship between the AI-TODv1.5 dataset and the VisDrone2019 dataset? Why is the AI-TODv1.5 dataset used for ablation experiments? How does this ablation experiment validate the effectiveness of the model?

Response 2: Thank you for pointing this out. The AI-TODv1.5 dataset and the VisDrone2019 dataset are both specialized datasets for small object detection. However, compared to the VisDrone2019 dataset, the objects to be detected in the AI-TODv1.5 dataset occupy a smaller proportion of the entire image. Additionally, the images in the AI-TODv1.5 dataset are captured from higher altitudes and include many examples resembling remote sensing images, making detection more challenging. The AI-TODv1.5 dataset poses a challenge for all object detectors. If our method can yield satisfactory results on this dataset, it would demonstrate the effectiveness of our approach for object detection from a drone's perspective.

Comments 3: Section 4.4 conducts comparative tests based on the PascalVOC dataset to ensure greater reliability and accuracy. It is recommended that the models compared be consistent with those in Table 1.

Response 3: Thank you for pointing this out. We have adjusted the comparative models in Table 2 to make them as consistent as possible with the models in Table 1. The change can be found-page 16, line 582

Reviewer 2 Report

Comments and Suggestions for Authors

This study proposes a feature enhanced small object detection algorithm based on the YOLOv8s model, which has sufficient content. However, there are some key issues in the article that the author needs to address and respond.

1. The abstract lacks specific indicators that reflect the experimental results;

2. The abstract content is too long and should be concise;

3. The first paragraph of the introduction should appropriately cite several references;

4. There is a citation error in line 89;

5. Article 4 of the contribution is only a work and should not be listed as a contribution to this article;

6. In related work, the development history of YOLO should not be introduced in too much detail, but rather the work done by other researchers in the relevant field that is relevant to this article;

7. The position of Figure 1 is incorrect;

8. Please add an improved YOLOv8 model diagram;

9. The study conducted experiments using different datasets, and the experimental content was very comprehensive. However, the author should provide a more specific description of the significance of conducting experiments using each dataset;

10. Add a P-R curve for YOLOv8s and YOLOv8m to better compare with the improved model;

11. In Table 2, is the parameter count of YOLO11s higher than that of YOLOv8s? It is understood that YOLO11 has improved its detection capability and reduced parameters based on YOLOv8;

12. Explain the reason for the significant decrease in FPS while reducing the number of parameters in the improved model;

13. Discuss unexpected results or patterns that arise in the research.

Author Response

Comments 1: The abstract lacks specific indicators that reflect the experimental results.

Response 1: Thank you for pointing this out. I have already added descriptions of the experimental results in the abstract of my paper. The change can be found-page 1. 

Comments 2: The abstract content is too long and should be concise

Response 2: Thank you for pointing this out. I have simplified the abstract section of the paper. The change can be found-page 1. 

Comments 3: The first paragraph of the introduction should appropriately cite several references

Response 3: Thank you for pointing this out. I have cited some references in the first paragraph of the paper. The change can be found-page 1. 

Comments 4: There is a citation error in line 89

Response 4: Thank you for pointing this out. I have made the corrections. The change can be found-line 74. 

Comments 5: Article 4 of the contribution is only a work and should not be listed as a contribution to this article

Response 5: Thank you for pointing this out. I have removed the fourth contribution stated in the paper.

Comments 6: In related work, the development history of YOLO should not be introduced in too much detail, but rather the work done by other researchers in the relevant field that is relevant to this article

Response 6: Thank you for pointing this out. I have simplified the section on the YOLO series in the related work and also added a section on the detection head, which is relevant to the research content of this paper. The change can be found-page 3, line 137 and page 5,line 232. 

Comments 7: The position of Figure 1 is incorrect

Response 7: Thank you for pointing this out. I have made the corrections. The change can be found-page 7. 

Comments 8: Please add an improved YOLOv8 model diagram

Response 8: Figure 1 is the our proposed model diagram and it is the improved YOLOv8 model diagram.

Comments 9: The study conducted experiments using different datasets, and the experimental content was very comprehensive. However, the author should provide a more specific description of the significance of conducting experiments using each dataset;

Response 9: Thank you for pointing this out. I have followed your suggestion and added explanations for why this dataset was used for the comparative experiments or ablation studies. The change can be found-page 13.

Comments 10: Add a P-R curve for YOLOv8s and YOLOv8m to better compare with the improved model

Response 10: Thank you for pointing this out. I have added a P-R curve for YOLOv8s and YOLOv8m. The change can be found-page 14, line 537 and line 554-556.

Comments 11: In Table 2, is the parameter count of YOLO11s higher than that of YOLOv8s? It is understood that YOLO11 has improved its detection capability and reduced parameters based on YOLOv8;

Response 11: Thank you for pointing this out. This was my mistake; I incorrectly listed the disk space occupied by the weight file in the table, but I have corrected it now. The change can be found-page 16, line 582.

Comments 12: Explain the reason for the significant decrease in FPS while reducing the number of parameters in the improved model

Response 12: The reduction in the number of parameters in our proposed model is due to the addition of a detection head, which receives feature maps with relatively fewer channels. Since the number of parameters in YOLOv8 is closely related to the number of channels in the feature map entering the first detection head, this leads to a decrease in the overall parameter count of our final model. However, in terms of computational complexity, the addition of the detection head and the incorporation of the attention mechanism have significantly increased the computational load. FPS is not only related to the model's parameter count but is also closely tied to the model's computational complexity. As the computational load increases, more computational resources are required during inference, which leads to a decrease in the FPS of our final model.

Comments 13: Discuss unexpected results or patterns that arise in the research.

Response 13: Compared to YOLOv4 and YOLOv8m, these two models have significantly more parameters than the model we proposed. Additionally, our model's advantages were not fully realized on conventional datasets, resulting in slightly lower performance metrics compared to these two models. The change can be found-page 17, line 595.

Round 2

Reviewer 2 Report

Comments and Suggestions for Authors

The quality of the paper has been improved through revisions.